# Assessing the Potential Distribution of *Oxalis latifolia*, a Rapidly Spreading Weed, in East Asia under Global Climate Change

**DOI:** 10.3390/plants12183254

**Published:** 2023-09-13

**Authors:** Anil Poudel, Pradeep Adhikari, Chae Sun Na, June Wee, Do-Hun Lee, Yong Ho Lee, Sun Hee Hong

**Affiliations:** 1Department of Plant Resources and Landscape Architecture, College of Agriculture and Life Sciences, Hankyong National University, Anseong 17579, Republic of Korea; aneeelily@gmail.com; 2Institute of Humanities and Ecology Consensus Resilience Lab, Hankyong National University, Anseong 17579, Republic of Korea; pdp2042@gmail.com; 3Wild Plant Seed Division, Baekdudaegan National Arboretum, Bong Hwa 36209, Republic of Korea; chaesun.na@koagi.or.kr; 4OJeong Resilience Institute, Korea University, Seoul 02841, Republic of Korea; dnlwns@korea.ac.kr; 5National Institute of Ecology, Seocheon 33657, Republic of Korea; eco0407@nie.re.kr

**Keywords:** broadleaf woodsorrel, habitat suitability, invasive species, maximum entropy, species distribution models

## Abstract

*Oxalis latifolia*, a perennial herbaceous weed, is a highly invasive species that poses a threat to agricultural lands worldwide. East Asia is under a high risk of invasion of *O. latifolia* under global climate change. To evaluate this risk, we employed maximum entropy modeling considering two shared socio-economic pathways (SSP2-4.5 and SSP5-8.5). Currently, a small portion (8.02%) of East Asia is within the *O. latifolia* distribution, with the highest coverages in Chinese Taipei, China, and Japan (95.09%, 9.8%, and 0.24%, respectively). However, our projections indicated that this invasive weed will likely be introduced to South Korea and North Korea between 2041 and 2060 and 2081 and 2100, respectively. The species is expected to cover approximately 9.79% and 23.68% (SSP2-4.5) and 11.60% and 27.41% (SSP5-8.5) of the total land surface in East Asia by these time points, respectively. South Korea and Japan will be particularly susceptible, with *O. latifolia* potentially invading up to 80.73% of their territory by 2081–2100. Mongolia is projected to remain unaffected. This study underscores the urgent need for effective management strategies and careful planning to prevent the introduction and limit the expansion of *O. latifolia* in East Asian countries.

## 1. Introduction

*Oxalis latifolia* Kunth, commonly called “broadleaf woodsorrel”, is a perennial herbaceous plant native to Mexico and Central and South America. It was introduced to Africa, Asia, and Australasia before 1950 for ornamental purposes [1,2]. It is distributed throughout tropical, Mediterranean, and temperate climates [3]. It is a low-growing plant with three broad leaflets on long petioles and reproduces both vegetatively by means of bulbs and sexually by means of seeds [4,5].

*O. latifolia* is referred to as a noxious and invasive weed as it has caused losses of 30 different types of crops, including rice, tea, potato, and apple, in 37 countries [3,6,7]. Its uncontrolled growth has resulted in a 56% maize reduction in India [8]. *O. latifolia* absorbs a relatively high proportion of soil nutrients and moisture and creates an allelopathic effect, inhibiting the growth of surrounding plants [9]. The weed also has a significant impact on nurseries, resulting in a loss of garden plants.

The distribution and invasiveness of plants are strongly influenced by climatic components, including temperature, precipitation, and atmospheric carbon dioxide [10]. Temperature plays important roles in the activation of *O. latifolia* bulbs [3]. The annual cycle begins when the bulbs are activated, occurring at temperatures above 15 °C in soil, and dormancy can last for more than 1 year [6]. The dormancy of bulbs is broken by a number of circumstances, including chilling at 5 °C for 3 weeks or dry heat at 45 °C for several hours, and continues until autumn in severe droughts [11,12].

East Asian countries, including China, Chinese Taipei, Japan, South Korea, North Korea, and Mongolia, confront a significant threat from invasive plants. These intruders endanger biodiversity, ecosystems, and human well-being. China, Chinese Taipei, Japan, and South Korea, due to increased global trade and tourism, face heightened invasion risks. China’s diverse climates and habitats house 488 invasive species across various environments [13,14]. Chinese Taipei, with tropical landscapes, notes 168 invasive plants, such as *O. latifolia*. Japan and South Korea record 154 and 320 invasive species, respectively, impacting agriculture and pastures [15,16]. Landlocked Mongolia, boasting diverse ecosystems, grapples with 77 invasive species listed in the Global Register of Introduced and Invasive Species (GRIIS), comprising 4.8% of its flora [17]. These invasions pose serious challenges across the region.

To manage invasive species and preserve native biodiversity, it is important to understand how invasive species are established and spread under current and future climate change [18]. In the past two decades, species distribution models (SDMs) have emerged as one of the most effective methods for investigating the impact of climate change on species habitat characteristics [19]. Among various SDMs, the maximum entropy method (MaxEnt) is the most frequently used machine learning technique for modeling species niches and geographic distributions using presence-only data based on parameters that link environmental variables and habitat suitability [20,21,22]. The MaxEnt algorithm has become an extremely popular tool for predicting the current and future potential habitats of invasive species because absence data are rarely available for these taxa and an equilibrium state has not been reached [23,24,25].

In East Asia, *O. latifolia* currently occurs in China and Chinese Taipei, with predictions of introduction from South Asia (e.g., India). It is well established in disturbed areas and is rapidly expanding into agriculture and pastures, resulting in productivity loss in such areas [3]. Future climate change is likely to favor the habitat expansion of *O. latifolia*, extending to other countries in East Asia. While the taxonomy, ecology, and physiology of *O. latifolia* have been studied [1,3,5,6], its distribution in relation to global climate change and human-induced disturbances has not been explored yet. Furthermore, understanding the potential distribution of *O. latifolia* is crucial for designing a robust quarantine system and implementing effective control and eradication measures.

Therefore, this study was designed with the main objectives as follows: (1) to identify important bioclimatic factors for the habitat suitability of *O. latifolia*, (2) to predict the potential distribution of *O. latifolia* in six East Asian countries using the MaxEnt modeling approach under the current and future climate change scenarios, and (3) to compare and estimate mean habitat suitability in six countries of East Asia under SSP2-4.5 and SSP5-8.5 scenarios. We expect the results to guide the development of a theoretical framework for preventing the introduction and dispersal of the species.

## 2. Results

### 2.1. Contribution of Bioclimatic Variables and Evaluation of Model Performance

We evaluated the degree of importance of various bioclimatic variables (Bio1, Bio2, Bio3, Bio12, Bio13, and Bio14) for modeling the distribution of *O. latifolia* based on the average contribution of each variable in the model for the time periods of 1970–2000, 2041–2060, and 2081–2100 (Table 1). Among six bioclimatic variables, three variables (i.e., Bio1, Bio3, and Bio12) had relatively high contributions to the model, estimated to be 35.23%, 30.08%, and 24.24%, respectively (Table 1). The distribution of *O. latifolia* was therefore primarily driven by annual mean temperature, isothermality, and annual precipitation, whereas other variables had relatively minor roles in model performance.

Variable importance was assessed using the jackknife method, which estimates the degree of relevance and distinctness of each variable to a species distribution model. Jackknife tests also revealed that Bio1, Bio3, and Bio12 were relatively important variables for predicting the potential distribution of *O. latifolia* (Appendix A).

In an evaluation of model performance, AUC, TSS, and Kappa values were 0.91, 0.85, and 0.71, respectively. The AUC value indicates the excellent model performance for predicting the global distribution of *O. latifolia*. Similarly, the evaluation metrics, TSS and Kappa, showed that there is good agreement in model predictions between observed and predicted data sets. These results confirm that the model exhibited excellent performance in predicting the spatial distribution of *O. latifolia* using presence-only data.

### 2.2. Predicting the Distribution of O. latifolia under Global Climate Change

Based on GBIF data, *O. latifolia* has been reported in three countries in East Asia, China, Japan, and Chinese Taipei, with no records in other countries, such as Mongolia, North Korea, and South Korea, in the wild (Figure 1).

MaxEnt modeling was performed to predict the potential distribution of *O. latifolia* in East Asian countries under current and future climate change scenarios (shared socio-economic pathways SSP2-4.5 and SSP5-8.5) and the results are presented in Figure 2. Currently, Chinese Taipei has highest proportion of suitable habitat relative to the total land surface of the country (95.09%), followed by China (9.78%) and Japan (0.24%) (Table 2). Mongolia, North Korea, and South Korea are climatically unsuitable for the establishment of *O. latifolia*. Altogether, 8.02% of the area of East Asia (689,716 cells, 2.5 min resolution) was estimated to be invaded by *O. latifolia* under the current climate.

Under future climate change scenarios, it was predicted that *O. latifolia* will be introduced to South Korea between 2041 and 2060. It was estimated that the species will cover approximately 9.82% and 15.89% of the total land surface under the SSP2-4.5 and SSP5-8.5 scenarios, respectively. Similarly, by 2081–2100, the suitable habitat for *O. latifolia* was predicted to expand by 77.29% (SSP2-4.5) and 80.73% (SSP5-8.5) across the country. In addition, *O. latifolia* was predicted to be introduced to North Korea by 2081–2100, while Mongolia was expected to remain safe from *O. latifolia* invasion at least until the end of this century. Other countries, such as China, Chinese Taipei, and Japan, had predicted invasion rates ranging from 27.24% to 99.45% (2041–2060) and 31.62% to 100% (2081–2100), as shown in Table 2. In total, by 2041–2060 and 2081–2100, approximately 9.79% and 23.68% (SSP2-4.5) and 11.60% and 27.41% (SSP5-8.5) of the total land surface of East Asia, respectively, could be invaded by *O. latifolia*. These results revealed that the habitat expansion of *O. latifolia* would be extensive in East Asia under global climate change.

### 2.3. Habitat Suitability Index and Future Potential Habitats in East Asia

Mean habitat suitability for *O. latifolia* was estimated under future climate change scenarios (SSP2-4.5 and SSP5-8.5) in six countries of East Asia, as summarized in Figure 3. The habitat suitability index was highest in North Korea (0.3) in 2081–2100, followed by China, Japan and South Korea, and Chinese Taipei (0.16, 0.14, 0.04, and 0.01, respectively). However, Mongolia was identified as an unsuitable habitat for *O. latifolia* by the end of this century. Countries with higher habitat suitability indexes in the future will have higher rates of greenhouse gas emissions and greater anthropogenic disturbances affecting natural resources.

## 3. Discussion

We assessed the risk of *O. latifolia* invasion under future climate change scenarios. We obtained several remarkable results as follows. First, the annual mean temperature was the most significant determinant of the global distribution of *O. latifolia* (Table 1). Second, the current potential distribution of *O. latifolia* was largely in China and Chinese Taipei; however, by 2081–2100, *O. latifolia* was predicted to expand to all countries in East Asia, except Mongolia (Figure 2). Third, based on the mean habitat suitability index, the rate of habitat expansion is predicted to be relatively high in North Korea, China, and Japan; these countries are at a high risk of invasion under the climate change scenario SSP5-8.5 (Figure 3).

Global climate change refers to the persistent alteration of worldwide climate patterns, including temperature, precipitation, and wind flows. These changes primarily result from the accumulation of greenhouse gases, such as carbon dioxide, methane, and nitrous oxide. These gases trap heat in the atmosphere, contributing to the Earth’s warming. Human activities, particularly the burning of fossil fuels, deforestation, industrial processes, and agriculture, have significantly elevated the levels of these greenhouse gases in the atmosphere. This has led to higher temperatures, rising sea levels, loss of biodiversity, and an increased risk of invasive alien plant species. Over the past century, the global temperature has risen by 0.78 °C. Predictions indicate that by 2100, relative to the average temperature of 1850–1880, global mean temperatures could rise by 1.5 °C under the SSP2-4.5 scenario and 5.2 °C under the SSP5-8.5 scenario [26]. Global climate change might exacerbate the risk of invasive alien plant species by causing damage to ecosystems and intensifying competition within native ecological systems due to higher levels of greenhouse gases [27]. To manage the invasion risk posed by these alien and invasive species, it is essential to anticipate their potential introduction, establishment, and spread, which can be achieved through standardized modeling systems.

SDM is widely used in ecological research, including the prediction of habitat suitability, assessment of biodiversity, and management of nature reserves [28,29,30]. More recently, SDM is used in the prediction of the biological invasion of alien plant species [31,32,33]. Several factors contribute to the predictive performance of SDM, including the size and resolution of the study area and the threshold used for modeling [34]. Other important factors include the type of species, number of species occurrence records, and environmental variables used in the models [35] as well as the modeling algorithms and global circulation models utilized [36] and the methods for model evaluation and validation [36,37].

In this study, we applied the MaxEnt modeling approach because the algorithm has high predictive performance using presence-only data and background data as pseudoabsence data [38]. Moreover, the algorithm can be run easily with many options in default settings [38]. Although MaxEnt is a powerful and widely used modeling technique, it is important to be aware of its limitations, such as the possibility of over-predicting species distributions due to sampling bias in geographic and environmental space [39]. To address this issue, we spatially filtered the species presence points in the model [40]. The scores for three indicators of predictive performance, AUC, TSS, and Kappa, were 0.91, 0.85, and 0.71, respectively, indicating that the model was robust and highly accurate in predicting the global distribution of the species.

Bioclimatic variables, such as temperature and precipitation, have a significant impact on the invasion of plant species [3,18,41]. Among the six bioclimatic varibles used to model the global, *O. latifolia* distribution, the temperature-related variable Bio1, and precipitation-related variable Bio12, were significant factors. These results indicate that *O. latifolia* requires a warm and humid environment for its optimal germination, growth, and spread. Therefore, it can survive in equatorial regions, including central America, equatorial South America, and introduced regions of Africa and Australia, with a temperature range of 20–25 °C and annual precipitation of 800–1500 mm [42]. However, in temperate and Mediterranean climates, aboveground vegetative parts disappear during the cold and wet autumn and reappear in late spring when temperatures increase [3,43]. In temperate and Mediterranean climates, the spring season is characterized by high levels of rainfall, resulting in activation of the bulbs. Thus, temperature may be a more significant factor than moisture for activation [43].

Globally, the native range of *O. latifolia* spans 19 countries in North, Central, and South America; however, it has been introduced to more than 41 countries of Asia, Europe, Australia, and Oceania [44]. *O. latifolia* has invaded some countries in South Asia (e.g., India, Nepal, and Bhutan) and Southeast Asia (e.g., China and Indonesia). However, it has not been recorded in South Korea, North Korea, or Mongolia. Our model predicted that these countries are climatically unsuitable for *O. latifolia* growth under the current climate. During the winter season, the average minimum temperature ranges between −2 °C and −20 °C in the southern coastal and northeastern regions of the Korean peninsula, respectively (KMA 2020). Similarly, the average minimum temperature in Mongolia ranges between −15 °C and −35 °C [45]. *O. latifolia* bulbs can survive short periods of freezing but are killed during exposure to sub-zero temperatures for prolonged periods [42]. Therefore, the climates of the northern part of China, North Korea, South Korea, and Mongolia are unsuitable under the current climate.

The annual mean temperature is expected to rise in many developed countries of East Asia, e.g., China, Japan, and South Korea, with estimated increases of 6.0 °C, 5.85 °C, and 5.38 °C, respectively, under SSP5-8.5 by 2100 [46,47,48], providing suitable climatic conditions for the introduction and habitat expansion of alien and invasive plants (e.g., *O. latifolia*) with tropical and subtropical origins [10]. Our model showed that *O. latifolia* would spread to South Korea, Japan, and China, covering approximately 9.82%, 5.4%, and 11.73% of the total area, respectively, by 2041–2060. Similarly, by 2081–2100, the habitat of *O. latifolia* was predicted to expand in northern regions of East Asia (up to 40° latitude), including North Korea, and spread to about 31.62% of China, 41.73% of Japan, and 77.29% of South Korea under SSP5-8.5. Mongolia is located above the 40° latitude and exhibits extreme cold in the winter, which does not allow the survival of warm-adapted plants, such as *O. latifolia.* Chinese Taipei is located in the southernmost part of East Asia, with a tropical and subtropical climate, favoring plant survival throughout the year under the current and future climates. In this study, we evaluated the habitat suitability index of *O. latifolia* in six East Asian countries to predict the rate of invasion under SSP2-4.5 and SSP5-8.5 considering anthropogenic activities, land cover changes, and population growth (and encroachment on natural resources) in 2081–2100. North Korea, China, and Japan had relatively higher values for the habitat suitability index and therefore will have higher rates of invasion under SSP5-8.5 than under SSP2-4.5; in these countries, the *O. latifolia* is expected to have weaker responses to greenhouse gas management efforts by 2081–2100 [49].

By 2081–2100, under the SSP2-4.5 and SSP5-8.5 climate change scenarios, agricultural land in the western and eastern regions of East Asia is at a high potential risk of *O. latifolia* invasion. This could result in significant economic losses and have a severe impact on food security, native biodiversity, and ecosystem services. Moreover, the invasion of *O. latifolia* degrades pastureland, severely affecting wild herbivores, including roe deer, sika deer, and ghorals [50]. Therefore, the control and management of *O. latifolia* are needed. Three major techniques, mechanical, chemical, and biological control, are widely used for invasive weeds [12]. Among these techniques, mechanical and chemical control are not effective for *O. latifolia* because they cannot destroy bulbs and roots in the soil [3,12]. Similarly, biological control could be an effective method for controlling *O. latifolia*, but it has not been studied or employed yet.

In addition to climate change, various other biotic and abiotic factors can accelerate the invasion of *O. latifolia* in East Asia (e.g., human activities, movement of goods, etc.) [51,52]. To improve the accuracy of predictions, it is crucial to incorporate various environmental variables into the model, including changes in land use and land cover, soil moisture, and soil pH, alongside bioclimatic variables. These are the limitations of our study, and we aim to use them in our next study. Looking ahead, we plan to evaluate the potential invasion risk of *O. latifolia* on a global scale using other environmental variables incluing bioclimatic variables. Additionally, we are focused on conducting invasion risk assessments tailored to specific countries in the near future.

## 4. Materials and Methods

### 4.1. Global Occurrence Points

Global occurrence data for *O. latifolia* (6406) were downloaded from the open access data portal, Global Biodiversity Information Facility (GBIF, www.gbif.org, accessed on 3 September 2022). To prevent the overfitting and incorrect inflation of model performance owing to spatial correlation [39,53], multiple species presence points within the same grid with a spatial resolution of 2.5 min were eliminated, leaving only a single unique point per grid by using the spatially rarefy occurrence tool in the ArcGIS SDM toolbox v. 2.4 [40]. The total number of *O. latifolia* occurrence points was reduced to 3595, as shown in Figure 1, and these points were used in the MaxEnt modeling (Appendix A) to predict the potential distribution of *O. latifolia* in East Asia, including China, Chinese Taipei, Japan, Mongolia, North Korea, and South Korea, under the current and future climate change scenarios SSP2-4.5 and SSP5-8.5.

### 4.2. Environmental Variables

Climatic variables are considered the most significant factors determining the potential distribution of a species [54]. Nineteen bioclimatic variables (Appendix A) considered important for predicting the *O. latifolia* distribution were downloaded from the World Climate (https://www.worldclim.org, accessed on 1 September 2022) [55] and were used to build models for predicting the species distribution at a global scale. Historical data for 1970–2000 were considered current climatic data and are referred to as across throughout the manuscript. Two shared socio-economic pathway (SSP) scenarios, SSP2-4.5 and SSP5-8.5, were employed to depict future climate data for the time periods 2041–2060 and 2081–2100. SSP scenarios describe future changes in socio-economic factors, such as human and economic factors, technology, land use, climate impact, emission of gases, and air pollutants [56,57]. It is predicted that the average global temperature will increase by 2.1–3.5 °C under SSP2-4.5 and 3.3–5.7 °C under SSP5-8.5 [58]. The bioclimatic variables were formed by the global climate model ‘Max Planck Institute for Meteorology Earth System Model (MPI-ESM1-2HR)’ [59] under the Coupled Model Intercomparison Project (CMIP6) [60] at a spatial resolution of 2.5 min (4.5 km at the equator).

The WorldClim portal is widely utilized for obtaining bioclimatic variables for predicting and investigating the potential distribution of species in response to changes in temperature and precipitation [61,62]. These variables helped to illustrate and predict the future distributions of species under various climate scenarios based on their ecologies [63]. Pearson correlation coefficients were evaluated to select bioclimatic variables that were not highly correlated with each other (r > 0.75, *p* = 0.05); highly correlated variables were removed to improve the predictive performance of the model [33,64]. Then, six bioclimatic variables, including temperature-related variables, e.g., annual mean temperature (Bio01), mean diurnal temperature range (Bio2), and isothermality (Bio03), and precipitation-related variables, e.g., annual precipitation (Bio12), precipitation in the wettest month (Bio13), and precipitation in the driest month (Bio14), were selected among the 19 bioclimatic variables based on the Pearson’s correlation coefficients (Appendix A) for modeling the *O. latifolia* distribution, as described previously [65]. The Pearson’s correlation analysis was performed using the PROC CORR function of SAS 9.4 (SAS Institute, Inc., Cary, NC, USA).

### 4.3. Model Development

The global distributions of *O. latifolia* under the current and future climates were predicted using MaxEnt Version 3.4.1 [66]. MaxEnt is an open-source, widely used machine learning technique that exhibits high predictive performance based on limited presence-only data sets [67,68]. It is the most appropriate tool for modeling invasive species because absence data are rarely available for these species and may not be accurate as their ranges may be expanding [69]. Therefore, we determined global background points using ArcGIS 10.8 (ESRI, Redlands, CA, USA) based on previously described methods [32,70]. To evaluate the predictive performance of the model, the species occurrence data were randomly split at a 3:1 ratio for model calibration and validation [28]. The model was replicated 100 times, and all other MaxEnt options were run with default parameters [31].

### 4.4. Model Evaluation and Validation

Model performance was evaluated using the area under the curve (AUC) of the receiver operator characteristic (ROC) curve (Pearson 2010), True Skill Statistic (TSS), and Kappa statistics [71]. The AUC serves as a threshold-independent method to evaluate model performance by separating presence from absence [72]. AUC values range from 0 to 1, where values of 0.5–0.6, 0.6–0.7, 0.7–0.8, 0.8–0.9, and 0.9–1 indicate fail, poor, fair, good, and excellent model performance, respectively [73]. The ROC curve analysis evaluates the prediction accuracy of the MaxEnt model based on AUC values. TSS is considered an alternative to AUC values for assessing model accuracy; it takes values ranging from −1 to +1 [71], with values closer to +1 indicating excellent sensitivity and values of zero or below denoting commission errors (i.e., expecting presence in the absence of the species) and omission (expecting absence in the presence of the species) [71,74]. Similarly, Kappa values range between −1 and +1, indicating poor agreement and perfect agreement, respectively. Furthermore, jackknife tests were performed to identify the important variables for predicting the potential spread of the target species [66].

### 4.5. Prediction of the Potential Habitat and Habitat Expansion of O. latifolia in East Asia

Global binary distribution maps of *O. latifolia* were determined using probability distribution maps produced from the MaxEnt model based on the maximum training sensitivity plus specificity Cloglog threshold. The binary distribution maps represent the presence and absence of *O. latifolia* habitats under the current and future climate change scenarios (SSP2-4.5 and SSP5-8.5) for the designated time periods, 1970–2000, 2041–2060, and 2081–2100. The binary distribution maps of East Asia for each time period were extracted using global binary distribution maps of *O. latifolia* by using the extract by mask option of the spatial analyst tool in ArcGIS Desktop 10.8. The numbers of suitable habitat cells under the climate change scenarios, SSP2-4.5 and SSP5-8.5, in each country of east Asia were estimated using zonal statistics of the spatial analyst tool in ArcGIS. Then mean habitat suitability of *O. latifolia* in each country of East Asia was estimated using number of suitable habitat cells to the total number of cells in such countries during the time period of 2081–2100. Habitat suitability was estimated using the ratio of mean habitat suitability under SSP2-4.5 to SSP5-8.5 subtracted from 1. This provides a basis for evaluating the rate of invasion of *O. latifolia* with respect to different SSP scenarios and thereby different greenhouse gas emissions, anthropogenic land cover changes, and human population growth rates.

## 5. Conclusions

*O. latifolia* is a noxious weed that threatens plant nurseries, gardens, and crop fields, with important economic consequences. Utilizing presence-only data for *O. latifolia* in East Asia and environmental variables, we employed MaxEnt modeling to determine the current and future distributions of the species. Our results indicate that, at present, the suitable habitat for *O. latifolia* is limited to a small portion of East Asia (approximately 8.02%), particularly in China, Chinese Taipei, and Japan. However, under global climate change, the coverage is projected to expand to approximately 27.41% of the total land surface of East Asia between 2081 and 2100. Additionally, mean habitat suitability was estimated for each country, revealing that North Korea, China, Japan, and South Korea would become future invasion hotspots for *O. latifolia*. Our study highlights the potential invasion risks posed by *O. latifolia* in East Asia. These results could be instrumental in planning preventive measures, such as strong quarantine measures to prevent the introduction and spread of this noxious weed. They can also aid in the development of long-term management strategies at both regional and local scales in East Asian countries.

## Figures and Tables

**Figure 1 plants-12-03254-f001:**
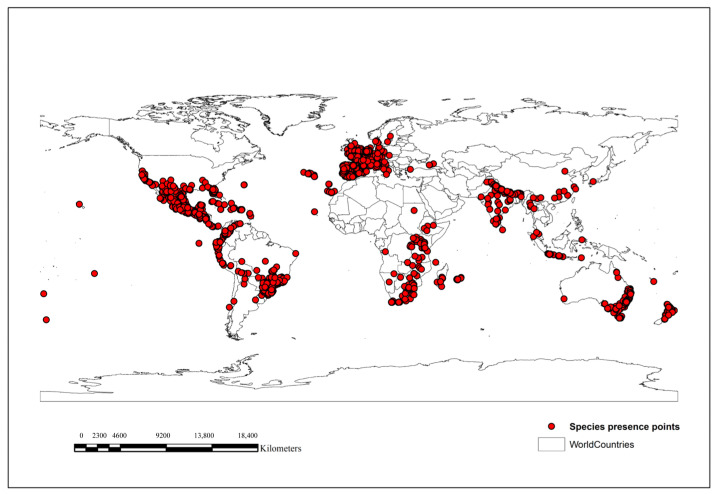
Worldwide occurrence records of *O. latifolia.* Red points indicate occurrence records.

**Figure 2 plants-12-03254-f002:**
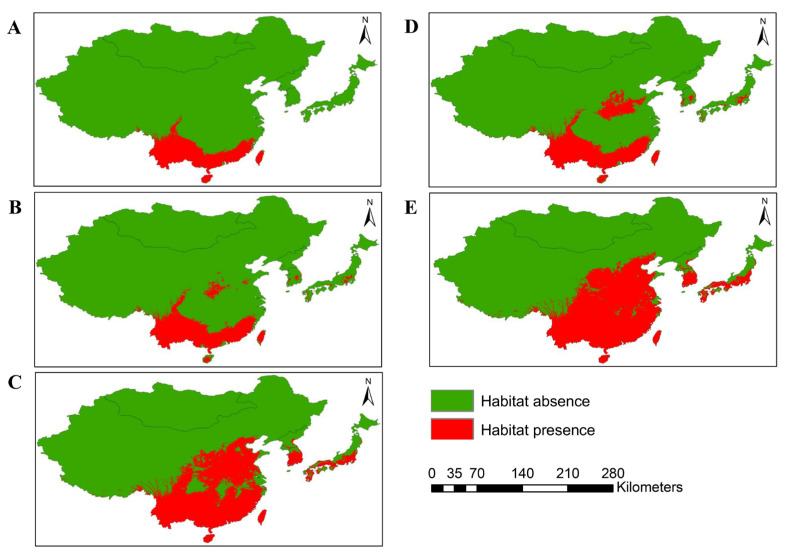
Distribution of *O. latifolia* in East Asia under current and future climatic scenarios: (**A**) habitat distribution of *O. latifolia* under the current climate (1970–2000); (**B**,**C**) potential distribution by 2041–2060 and 2081–2100 under the climate change scenario SSP2-4.5; (**D**,**E**) potential distribution by 2041–2060 and 2081–2100 under the climate change scenario SSP5-8.5. Green and red in the legend indicate the absence and presence of habitats of *O. latifolia*.

**Figure 3 plants-12-03254-f003:**
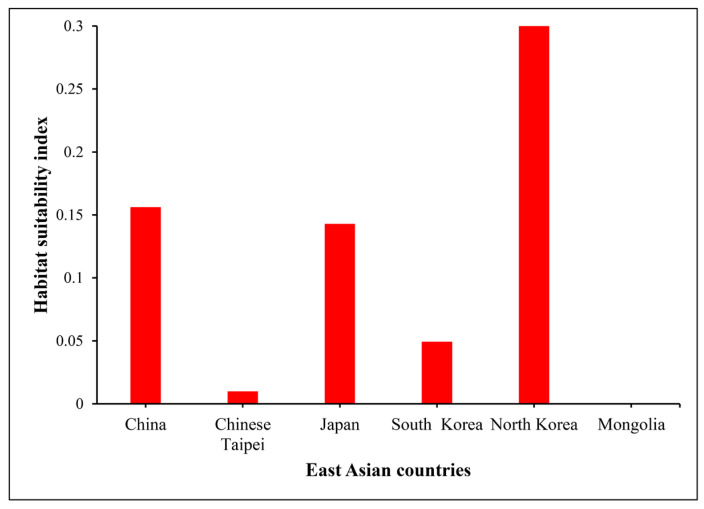
Graphical representation showing differences in the ratios of the mean habitat suitability index of *O. latifolia* between climate change scenarios SSP2-4.5 and SSP5-8.5 in various countries of East Asia by the period of year 2081–2100.

**Table 1 plants-12-03254-t001:** Selected bioclimatic variables and their average contribution to the model.

Variable	Variable Description	Units	Model Contribution (%) *
Bio1	Annual mean temperature	°C	35.23
Bio2	Mean diurnal temperature range	°C	1.46
Bio3	Isothermality (BIO2/BIO7) (×100)	%	30.08
Bio12	Mean annual precipitation	mm	24.24
Bio13	Precipitation of wettest month	mm	0.18
Bio14	Precipitation of driest month	mm	8.8

* Contribution of selected environmental variables in the MaxEnt model for *O. latifolia* under the current and future climatic scenarios, SSP2-4.5 and SSP5-8.5, for the time periods of 1970–2000, 2041–2060, and 2081–2100.

**Table 2 plants-12-03254-t002:** Percentage of area covered by *O. latifolia* in different countries in East Asia under the climate change scenarios SSP2-4.5 and SSP5-8.5.

Countries	Total		SSP2-4.5 (%)		SSP5-8.5 (%)	
	Cell Number	1970–2000	2041–2060	2081–2100	2041–2060	2081–2100
China	547,295	9.78	11.73	27.24	13.89	31.62
Chinese Taipei	1832	95.09	88.05	99.45	90.28	100
Japan	21,281	0.24	5.4	35.75	6.95	41.73
South Korea	5589	0	9.82	77.29	15.89	80.73
North Korea	7454	0	0	7.24	0	10.25
Mongolia	106,265	0	0	0	0	0
Total ^a^	689,716	8.02	9.79	23.68	11.60	27.41

^a^ Total cell numbers in six countries of East Asia (China, Chinese Taipei, Japan, South Korea, North Korea, and Mongolia). Percentage indicates the proportion of cells predicted to have suitable habitats relative to the total number of cells for each country. The cell of each raster was 2.5 min or 4.5 km at the equator. SSP, shared socio-economic pathway.

## Data Availability

Not applicable.

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
