# Peer review of "Assessing the Potential Distribution of *Oxalis latifolia*, a Rapidly Spreading Weed, in East Asia under Global Climate Change"

_plants, 2023, doi:10.3390/plants12183254_

Round 1
Reviewer 1 Report
Overall the study is interesting and well written. The topic is relevant and the outcomes of the study are important for future management plans of invasive alien species and weeds. Chosen species is even more relevant as it could be a noxious crop weed.
The study contains all crucial information: background and the motivation for the study, specific aims, methodology, research outcome and discussion of the results. However, the introduction and methods description could be improved. Specific comments are in the attachment.
Author Response
Reviewer 1
Overall the study is interesting and well written. The topic is relevant and the outcomes of the study are important for future management plans of invasive alien species and weeds. Chosen species is even more relevant as it could be a noxious crop weed. The study contains all crucial information: background and the motivation for the study, specific aims, methodology, research outcome and discussion of the results. However, the introduction and methods description could be improved. Specific comments are in the attachment.
Response: We are grateful for your valuable suggestions and nice comments. As suggested, we have improved the description of introduction section line no. 51-88 and corrected minor typing mistakes in the material and methods section.
Reviewer 2 Report
In the manuscript “Assessing the potential distribution of Oxalis latifolia, a rapidly spreading weed, in East Asia under global climate change” the authors e employed Maximum Entropy modeling considering two Shared Socio-economic Pathways (SSP2-4.5 and SSP5-8.5). This manuscript is well organized, and the drawn conclusions are coherent with the obtained results. The paper was well written!
Lines 29 – 30: Please, arrange the keywords alphabetically.
Bertolino, S., et al., (2020). Spatially explicit models as tools for implementing effective management strategies for invasive alien mammals. Mammal Review, 50(2), 187-199.
Line 53: It is “Species Distribution Models”.
Lines 55 – 58: I think that you should these important references as examples to support your sentence: “Among various species distribution models (SDMs), the maximum entropy method (MaxEnt) is the most frequently used machine learning technique for modeling species niches and geographic distributions using presence-only data based on parameters that link environmental variables and habitat suitability”. I would like to suggest:
Bosso, L., et al., (2017). Modelling the risk posed by the zebra mussel Dreissena polymorpha: Italy as a case study. Environmental Management, 60, 304-313.
Rahimian Boogar, A., et al., (2019). Predicting habitat suitability and conserving Juniperus spp. habitat using SVM and maximum entropy machine learning techniques. Water, 11(10), 2049.
Sharifian, S., et al., (2023). Predicting present spatial distribution and habitat preferences of commercial fishes using a maximum entropy approach. Environmental Science and Pollution Research, 1-14.
Lines 59 – 61: I think that you should an important reference as example to support your sentence: “The MaxEnt algorithm has become an extremely popular tool for predicting the current and future potential habitats of invasive species because absence data are rarely available for these taxa and an equilibrium state has not been reached”. I would like to suggest:
Raffini, F., et la., (2020). From nucleotides to satellite imagery: Approaches to identify and manage the invasive pathogen Xylella fastidiosa and its insect vectors in Europe. Sustainability, 12(11), 4508.
Lines 72 – 79: Please, highlight better your predictions and hypothesis.
Lines 135: Please, add the north symbol in the figure.
Lines 160 – 267: Please, expand this section of the manuscript.
Lines 167 – 168: Please, move this figure in the results.
Line 270 – 279: Did you analyse your presence records for spatial autocorrelation?
Line 274: What is the spatial resolution of your points?
Lines 299 – 301: I think that you should these important references as example to support your sentence: “The WorldClim portal is widely utilized for obtaining bioclimatic variables for predicting and investigating the potential distribution of species in response to changes in temperature and precipitation”. I would like to suggest:
Anand, V., et al., (2021). Predicting the current and future potential spatial distribution of endangered Rucervus eldii eldii (Sangai) using MaxEnt model. Environmental Monitoring and Assessment, 193(3), 147.
Bosso, L., et al., (2022). The rise and fall of an alien: Why the successful colonizer Littorina saxatilis failed to invade the Mediterranean Sea. Biological Invasions, 24(10), 3169-3187.
Line 316 – 325: Please, add all the information of the setting used in Maxent in your modelling analysis. Please, add all the information in the supplementary materials.
Lines 364 – 377: Please, add some information about potential future study starting from the results of this paper.
Extensive editing of English language required
Author Response
Reviewer 2
In the manuscript “Assessing the potential distribution of Oxalis latifolia, a rapidly spreading weed, in East Asia under global climate change” the authors e employed Maximum Entropy modeling considering two Shared Socio-economic Pathways (SSP2-4.5 and SSP5-8.5). This manuscript is well organized, and the drawn conclusions are coherent with the obtained results. The paper was well written!
Response: Thank you for your valuable comments.
Lines 29 – 30: Please, arrange the keywords alphabetically.
Response: As suggested, we arranged all keywords alphabetically.
Bertolino, S., et al., (2020). Spatially explicit models as tools for implementing effective management strategies for invasive alien mammals. Mammal Review, 50(2), 187-199.
Response: We have read this article for understanding the management strategies of alien mammals. But we cannot find citation materials from this article.
Line 53: It is “Species Distribution Models”.
Response: As suggested, we have corrected it. (Line no. 64)
Lines 55 – 58: I think that you should these important references as examples to support your sentence: “Among various species distribution models (SDMs), the maximum entropy method (MaxEnt) is the most frequently used machine learning technique for modeling species niches and geographic distributions using presence-only data based on parameters that link environmental variables and habitat suitability”. I would like to suggest:
Bosso, L., et al., (2017). Modelling the risk posed by the zebra mussel Dreissena polymorpha: Italy as a case study. Environmental Management, 60, 304-313.
Response: As suggested, we have read this article and cited in our revised manuscript (Line no. 460)
Rahimian Boogar, A., et al., (2019). Predicting habitat suitability and conserving Juniperus spp. habitat using SVM and maximum entropy machine learning techniques. Water, 11(10), 2019.
Response: As suggested, we have read this article and cited in our revised manuscript (Line no. 458)
Sharifian, S., et al., (2023). Predicting present spatial distribution and habitat preferences of commercial fishes using a maximum entropy approach. Environmental Science and Pollution Research, 1-14.
Response: As suggested, we have read this article and cited it in our revised manuscript (Line no. 462)
Lines 59 – 61: I think that you should an important reference as example to support your sentence: “The MaxEnt algorithm has become an extremely popular tool for predicting the current and future potential habitats of invasive species because absence data are rarely available for these taxa and an equilibrium state has not been reached”. I would like to suggest:
Raffini, F., et la., (2020). From nucleotides to satellite imagery: Approaches to identify and manage the invasive pathogen Xylella fastidiosa and its insect vectors in Europe. Sustainability, 12(11), 4508.
Response: As suggested, we have read this article and cited in our revised manuscript (Line no. 470)
Lines 72 – 79: Please, highlight better your predictions and hypothesis.
Response: As suggested, we have revised our manuscript. (Line no. 73-88)
Lines 135: Please, add the north symbol in the figure.
Response: As suggested, we have added North pole symbol in the figure. (Line no. 144)
Lines 160 – 267: Please, expand this section of the manuscript.
Response: As suggested, we have expanded the discussion section of the manuscript (Line no. 174-295)
Lines 167 – 168: Please, move this figure in the results.
Response: As suggested, we have moved the figure no. 3 (line no. 168).
Line 270 – 279: Did you analyse your presence records for spatial autocorrelation?
Response: No, we have not analyzed presence records for spatial autocorrelation. However, we performed spatial rarifying using the SDM toolbox in Arc GIS to prevent potential spatial autocorrelation.
Line 274: What is the spatial resolution of your points?
Response: Spatial resolution of our points is 2.5 min.
Lines 299 – 301: I think that you should these important references as example to support your sentence: “The WorldClim portal is widely utilized for obtaining bioclimatic variables for predicting and investigating the potential distribution of species in response to changes in temperature and precipitation”. I would like to suggest:
Anand, V., et al., (2021). Predicting the current and future potential spatial distribution of endangered Rucervus eldii eldii (Sangai) using MaxEnt model. Environmental Monitoring and Assessment, 193(3), 147.
Bosso, L., et al., (2022). The rise and fall of an alien: Why the successful colonizer Littorina saxatilis failed to invade the Mediterranean Sea. Biological Invasions, 24(10), 3169-3187.
Response: As suggested we have added both references in our revised manuscript. (Line no. 547-551).
Line 316 – 325: Please, add all the information of the setting used in Maxent in your modelling analysis. Please, add all the information in the supplementary materials.
Response: As suggested we have provided inputs of Maxent model in supplementary materials.
Lines 364 – 377: Please, add some information about potential future study starting from the results of this paper.
Response: As suggested, we have provided future direction of our study in discussion section line no 291-295.
Comments on the Quality of English Language
Response: We have edited English language of this manuscript using a professional company Bioedit English editing service (Company no. 04150179, UK). The certificate of English edition is uploaded in the system.
Please see the attachment.

Reviewer 3 Report
In this manuscript (plants-2571477) entitled "Assessing the potential distribution of Oxalis latifolia, a rapidly spreading weed, in East Asia under global climate change" submitted to Plants, Anil Poudel and colleagues have employed Maximum Entropy modeling to evaluate the risk of invasion of Oxalis latifolia in East Asia under global climate change. This study underscores the urgent need for effective management strategies and careful planning to prevent the introduction and limit the expansion of O. latifolia in East Asian countries. This research is interesting, but the present manuscript is unsuitable for publication.
1. The formula employed in the modeling should be described in details in the revised manuscript.
2. For Figure 2, climate changes should be included in the revision.
3. For Table 2, what is the situation for the period of 2000-2041?
4. For Figure 3, analysis in significance of difference should be performed. Please label the significance of difference in the revised figure 3.
5. Taiwan is a region of China, not an independent country in East Asian. Authors should carefully correct this mistake throughout the manuscript.
Author Response
Reviewer 3
In this manuscript (plants-2571477) entitled "Assessing the potential distribution of Oxalis latifolia, a rapidly spreading weed, in East Asia under global climate change" submitted to Plants, Anil Poudel and colleagues have employed Maximum Entropy modeling to evaluate the risk of invasion of Oxalis latifolia in East Asia under global climate change. This study underscores the urgent need for effective management strategies and careful planning to prevent the introduction and limit the expansion of O. latifolia in East Asian countries. This research is interesting, but the present manuscript is unsuitable for publication.
- The formula employed in the modeling should be described in detail in the revised manuscript.
Response: Thank you for the suggestion. However, we were unable to provide details about the theoretical background of the modeling process. Our predictions were generated using an established model, utilizing a freely available package in the R script.
- For Figure 2, climate change should be included in the revision.
Response: As suggested climate change is included in Figure 2.
- For Table 2, what is the situation for the period of 2000-2041?
Response: We have not analyzed for this time period.
- For Figure 3, analysis of the significance of difference should be performed. Please label the significance of difference in the revised figure 3.
Response: Thank you for the suggestion but we have no replicated data sets so we could not analyze the significance difference for this analysis. We just estimated the ratio of two climate change scenarios SSP-2.45 and SSP5-8.5.
- Taiwan is a region of China, not an independent country in East Asian. Authors should carefully correct this mistake throughout the manuscript.
Response: As suggested, we have changed its name to Chinese Taipei, which is globally used in practice.
Round 2
Reviewer 2 Report
Well done!
Ok
Author Response
We are grateful to the editor and all reviewers for their invaluable comments and suggestions for improving our manuscript. We have meticulously checked all comments and suggestions and revised the manuscript accordingly.
